# Differences and Similarities between the Wood of Three Low-Density and Homogenous Species: Linden, Balsa, and Paulownia

Anamaria Avram, Aurel Lunguleasa *, Cosmin Spirchez  and Constantin Stefan Ionescu

Department of Wood Processing and Design of Wooden Products, Transilvania University of Brasov, 500036 Brasov, Romania; anamaria.avram@unitbv.ro (A.A.); cosmin.spirchez@unitbv.ro (C.S.); ionescu.constantin.stefan@unitbv.ro (C.S.I.)
* Correspondence: lunga@unitbv.ro

**Abstract:** The use of woods with a low density and good structural uniformity has intensified in recent decades, paving new ways for their superior valorization. This research study aimed to examine the wood of three deciduous wood species with low densities and great uniformity of structure, namely linden wood, with an oven-dried density of 461 kg/m$^3$, paulownia wood, with an oven-dried density of 304 kg/m$^3$, and balsa wood, with an oven-dried density of 118 kg/m$^3$. The physical and mechanical properties of these species were studied using various methodologies. The obtained results show that, although they have significant differences in terms of densities and mechanical characteristics, the coloristic properties of the three analyzed species obtained using the CIELab are almost similar. As a general conclusion, based on all the properties found in this research, it can be concluded that paulownia wood is more appropriate than linden or balsa for use, being able to replace linden in its uses.

**Keywords:** linden; paulownia; balsa; low density; structural uniformity

## 1. Introduction

The wood of different wood species differs greatly from one species to another. Even within the same genus and family, there is a diversity of different wood species. Going further, even within the same wood species, its wood can differ depending on the type of soil, climatic conditions, geographical area, etc. For the majority of wood-processing technologies, and in finished or semifinished products, a homogeneity of the wood species used is needed [1,2]; otherwise, the produced products will have major deficiencies or will have reduced marketability. This homogeneity can be characterized from a structural point of view, including density [3,4], reaction to atmospheric humidity or water, color [5] or the color difference between the sapwood and heartwood (mature wood) [6], the presence/absence of defects, resistances, and surface roughness [3], with all of these depending on the processing technology used and the final product that is obtained. It is known that the wood of the same wood species has a different structure. There are differences between the sapwood and heartwood, the annual rings of early and late wood, or the type of deciduous wood species. From this point of view, the most uniform species are those with small and uniform distributed pores, such as linden wood. There are fields in which only homogeneous woods are used due to variation in their density and structure [1,7], such as when casting molds in the shoe industry, in the textile industry, and in manual sculptures, and when casting molds of alloys of small shapes in the foundry industry.

In the field of manufacturing wood-based composite boards, such as LVL [8], wood species with appropriate densities are needed; only in this way can chips be compatible with and compacted according to the technology used in this field. Also, the color of the wood species used in mixture chips, flakes, or chops must be similar and uniform; otherwise, the

boards obtained will have uneven colors or a reduced esthetic appearance. In the field of restoration of cultural heritage objects with wooden supports, it is recommended to use the same wood species, especially species of a density equivalent to that found in the degraded object to be restored [9–11]. Historically, cultural heritage objects were mainly made of linden wood due to the structural homogeneity of this species, as well as the homogeneity of its shrinkage/swelling in the two structural directions (the ratio of tangential shrinkage to radial shrinkage). When it is restored, the wood in degraded heritage objects has a low density, with the mass losses being 25–50% depending on the degree of degradation. This is why the replaced wood material must have a density equivalent to that of the object to be restored (much smaller than the initial one); otherwise, the product would be heavier and additional stresses would be induced. Consequently, if linden wood is used for restoration, it should be artificially degraded or torrefied to reduce its density to that appropriate for the wood in the restored object, or to use another species with a lower density, such as balsa wood or paulownia wood [12,13]. In the field of veneers and timbers, species that are very closed in terms of density, texture, pattern, and color are considered uniform, even with similarities inside the annual rings [4,8]. The investigation methods of wood homogeneity were different from one author to another, and included the use of X-rays [4] or a combination of light transmittance and mechanical stiffness [8]. The analyzed materials included bamboo veneers and LVL [8], white fir, Aleppo pine, European beech, walnut, and white oak [3].

**Objectives**. This paper aims to analyze the uniformity of the wood of three homogeneous wood species, namely linden, balsa, and paulownia, from a structural and coloristic point of view, with respect to their physical–mechanical properties, and their relationship with water (water absorption and shrinkage). In this study, linden wood is taken as the element of comparison, emphasizing the specific properties of paulownia wood, which is promoted as the replacement for linden wood in all possible uses. This paper begins with the premise that linden and balsa woods are unsuitable species for replacing very old wood from heritage objects, and that paulownia wood could be a much more advantageous solution.

## 2. Materials and Methods

**Anatomical structure**. The first element of the analysis of the homogeneity or differences between the three wood species (linden, balsa, and paulownia) was their anatomical structure, including a study of the cross-sections of these three deciduous species, and their inclusion in the category of species with typically annular or uniformly scattered pores [1,6,7]. For this purpose, three cross-sections of 20 mm × 20 mm of the analyzed species were taken, as shown in Figure 1.

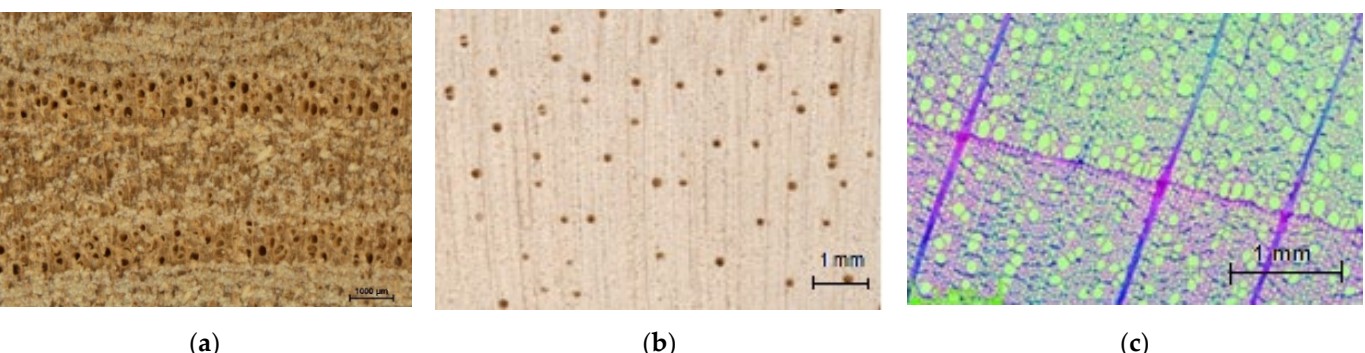

(**a**)      (**b**)      (**c**)

**Figure 1.** Microscopic cross-sectional images of paulownia (**a**), balsa (**b**), and linden (**c**).

**Unit density.** The density of the three wood species was determined as mass per unit volume, as defined by international standards in the field [14,15]. Because wood is found at different moisture content levels, for a good comparison of the three wood species, both the mass and the volume of 30 mm × 20 mm × 20 mm samples were determined at the same moisture content of 0% (oven-dried state). For this purpose, the samples were dried

for 24 hours in an oven with a 105 °C temperature. Taking into account the volume of the rectangular parallelepiped, under which the samples' density was determined, the wood density was calculated as follows (Equation (1)):

$$\rho_0 = \frac{m}{l \cdot b \cdot g} \cdot 10^6 \left[ \text{kg/m}^3 \right] \tag{1}$$

where $\rho_0$ is the density at a wood moisture content of 0%; $m$ is the mass of the wooden specimen, in $g$; $l$ is the length of the specimen, in mm; $b$ is the width of the specimen, in mm; and $g$ is the thickness of the specimen, in mm.

Ten samples were used for this test.

**The CIEL\*a\*b\* color of the 3 species**. Immediately after cutting its log into lumber, paulownia wood darkens in color to a light gray. This coloring is observable on a small thickness of about 0.7–0.8 mm due to the oxidation of wood sap when it comes into contact with atmospheric air [16–18]. This blackish color is kept as long as the wood lumber is not processed. The dark color will disappear when the timber surface is planned to a thickness of at least 0.7–0.8 mm, obtaining the real and beautiful color of paulownia wood surface [19–21].

A portable colorimeter of type Tes-135A, manufactured by Tess Electrical, Electronic Corp. (Taipei, Taiwan), was used for the color determination, using the CIEL\*a\*b\* system. The CIEL\*a\*b\* color space is one of the multiple color spaces (alongside CIEXYZ, CMYK, NCS, RGB, etc.) that was defined by the International Commission on Illumination (abbreviated as CIE) in 1976, which makes possible the transition from visual appreciation of colors to numerical quantification. It is based on human perception of colors as opposing colors, in which the group of red with green and the group of blue with yellow are usually opposing colors (Figure 2).

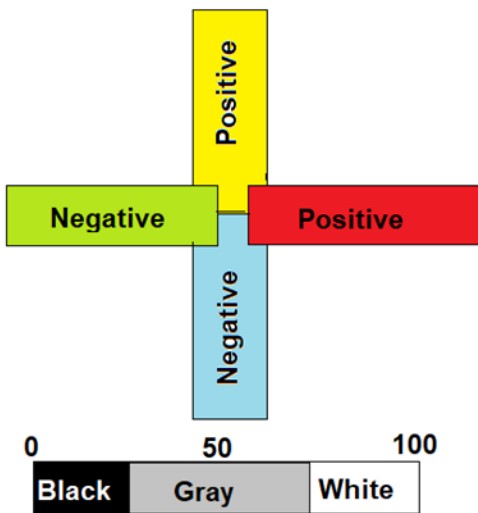

**Figure 2.** Coloristic space CIEL\*a\*b\*.

This is why the CIEL\*a\*b\* color space is quantified by three distinct parameters: L\*, a\*, and b\*. In this color space, the letter "L\*" represents lightness and has a value of 0 for black and 100 for white, with values in between being a series of shades of gray. The "a\*" axis refers to the green–red opposition, with negative values toward green and positive values toward red. The "b\*" axis quantifies the blue–yellow opposition, with negative values toward the blue zone and positive values toward the yellow zone [5,18].

For the color study, the three radial sections shown below were used (Figure 3).

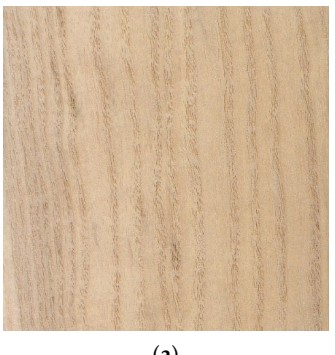 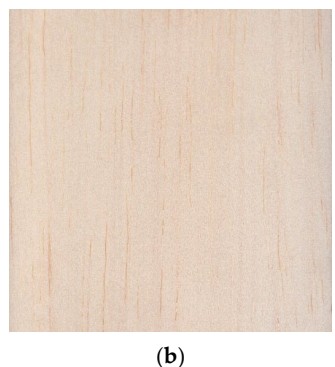 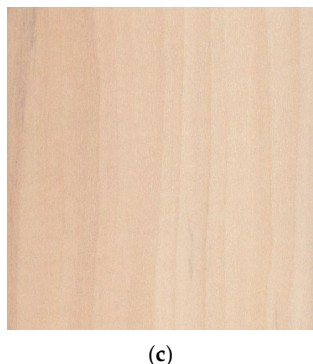

(**a**)                    (**b**)                    (**c**)

**Figure 3.** Radial sections for color determination: (**a**) *Paulownia tomentosa*; (**b**) balsa (*Ochroma pyramidale*); and (**c**) linden (*Tilia platyphyllos*).

From Figure 3, it can be seen visually that, roughly, the three species are similar in color, with paulownia wood being a little toward brown, and linden wood being a little toward reddish.

**Water absorption after two hours of immersion.** Water absorption after immersion in water for 2 h was determined for the absolutely dry samples, which were kept in the oven at 105 °C for 6 h [22–25]. The linden, balsa, and paulownia samples had the average dimensions of 20 mm × 20 mm × 30 mm, where the dimension of 30 mm was their length. The mass after immersion and the mass after drying in the oven were determined for the three wood species. Based on these masses, water absorption (*WA*) was determined according to the following relationship (Equation (2)):

$$WA = \frac{m_w - m_0}{m_0} \cdot 100 \; [\%] \tag{2}$$

where $m_w$ is the mass of the immersed specimen, in g, and $m_0$ is the mass of the oven-dried test piece, in g.

Seven specimens were used for this test.

**Shrinkage in the radial/tangential direction and volume.** Wood shrinkage was performed on the samples of the same types and dimensions as in the case of absorption. These samples were cut in such a way so that the radial and tangential directions were clearly identified [26,27]. In this test, with the help of an electronic caliper, the initial dimensions in the wet state and the final ones in the absolutely dry state were determined. Based on the masses of the samples in the wet and dry state, the wood moisture content was also determined.verifying whether the requirement of a certain moisture content above the fiber saturation point (over 30% moisture content) was fulfilled. If the moisture content of the samples did not touch 30% (Figure 4), the necessary correction was performed; the value for 1% moisture content was determined, and then, this value was multiplied by 30%.

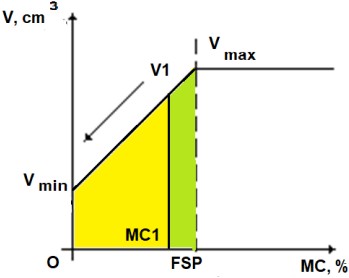

**Figure 4.** Wood shrinkage diagram: MC—moisture content, in %; FSP—fiber saturation point, in %; V$_{max}$—the maximum volume of wood, in cm³; V$_{min}$—the minimum volume of wood, in cm³; MC1—moisture content under FSP, in %; V1—volume of wood for moisture content MC1, in cm³.

The total shrinkage (throughout the saturation moisture range) in the radial, tangential, longitudinal, and volumetric directions was determined based on the dimensions before and after drying according to the following specific relationships (Equations (3) and (4)):

$$S_{tl} = \frac{l_{max} - l_{min}}{l_{max}} \cdot 100 \ [\%]; S_{tt} = \frac{t_{max} - t_{min}}{t_{max}} \cdot 100 \ [\%] \tag{3}$$

$$S_{tr} = \frac{r_{max} - r_{min}}{r_{max}} \cdot 100 \ [\%] \ S_{tv} = \frac{V_{max} - V_{min}}{V_{max}} \cdot 100 \ [\%] \tag{4}$$

where $S_{tl}$ is the total longitudinal shrinkage, in %; $l_{max}$ is the maximum length of the specimen, after immersion in water, in mm; $l_{min}$ is the minimum length of the sample, after drying in the oven, in mm; $S_{tt}$ is the total tangential shrinkage, in %; $t_{max}$ is the maximum thickness (on thickness) of the specimen, after immersion in water, in mm; $t_{min}$ is the minimum thickness of the sample, after drying in the oven, in mm; $S_{tr}$ is the total radial shrinkage, in %; $r_{max}$ is the maximum radial of the specimen, after immersion in water, in mm; $r_{min}$ is the minimum radial of the sample, after drying in the oven, in mm; $S_{tv}$ is the total volumetric shrinkage, in %; $V_{max}$ is the maximum volume of the specimen, after immersion in water, in mm; and $l_{min}$ is the minimum volume of the sample, after drying in the oven, in mm.

The partial volumetric shrinkage (only in a portion of the moisture fiber saturation), such as from the saturation point of the fiber to a certain MC1 value (Figure 4), can be determined according to the following equation (Equation (5)):

$$S_{pv} = \frac{V_{max} - V_1}{V_{max}} \cdot 100 \ [\%] \tag{5}$$

where $S_{pv}$ is the partial volumetric shrinkage, in %, and $V_1$ is the volume of wood corresponding to the MC1.

**Modulus of resistance and modulus of elasticity to static bending conforming to ISO 13061-4: 2014** [28]. Before testing, the samples were conditioned to obtain an average moisture content of 12%. The modulus of resistance (MOR) and modulus of elasticity (MOE) to static bending were determined according to the following general relationships (Equations (6) and (7)):

$$\sigma_i = \frac{3 \cdot F_{max} \cdot l}{2 \cdot b \cdot t^2} \left[ \frac{N}{mm^2} \right] \tag{6}$$

$$E_m = \frac{l_1^3 \cdot (F_1 - F_2)}{4 \cdot b \cdot t^3 \cdot (a_2 - a_1)} \left[ \frac{N}{mm^2} \right] \tag{7}$$

where $F_{max}$ is the maximum force of breaking, in N; $l$ is the distance between supports, in mm; $b$ is the width of the specimen, in mm; $t$ is the thickness of the specimen, in mm; $F_1$ is the value of force at 10% from the maximum force, in N; $F_2$ is the value of force at 40% from the maximum force, in N; $a_1$ is deformation for force $F_1$, in mm; and $a_2$ is deformation for force $F_2$.

Ten samples were tested in order to obtain statistically significant values.

**Compressive strength conforming to ISO 13061-17** [28,29]. Before testing, the samples were conditioned in order to have up to 12% moisture content. The compressive strength parallel to the grain used in the work was obtained in the moment of breaking when the layers of wood in the samples were sheared or crushed, and the force dropped suddenly.

Therefore, the resistance to compression parallel to the wood grain was determined as the ratio between the maximum breaking force and the area of the breaking surface (Equation (8)):

$$\sigma_c = \frac{F_{max}}{b_1 \times b_2} \left[ \frac{N}{mm^2} \right] \tag{8}$$

where $\sigma_c$ is the compressive strength, in $\text{N/mm}^2$; $F_{max}$ is the maximum force of breakage, in N; and $b_1$ and $b_2$ are the widths of the specimen in tangential and radial directions, in mm.

Ten replicates of each type of specimen were used for this test.

**Brinell hardness conforming to EN 1534:2003** [30]. Brinell hardness quantifies the resistance of a wood surface when a metal ball tries to penetrate and destroy it. The punch that is pressed onto the surface of the wood is created by a ball with a diameter of 10 mm, and, to highlight the diameter of the mark, it is necessary to arrange a copy paper between the punch and the wooden surface. This resistance is higher on the transverse section, compared to the longitudinal–radial and longitudinal–tangential sections, but, due to the fact that the transverse section is less used in industrial practice of wood processing, only the radial and tangential sections are used in research. Brinell hardness was calculated as follows (Equation (9)):

$$HB = \frac{2 \times P}{\pi \times D(D - \sqrt{D^2 - d^2})} \left[\frac{\text{N}}{\text{mm}^2}\right] \tag{9}$$

where $P$ is the applied force, related to wood density, in N; $D$ is the diameter of the punch ball, in mm; and $d$ is the diameter of the trace left on the wood, in mm.

Ten valid samples were used for testing each wood species.

**Ecological aspects of the three species.** All forests, through the trees that they have in their area, have several functions, among which are hydrological, soil protection, esthetic, recreational, ecological, and production functions [31]. The ecological function of forests is related to physiology, namely the fact that humans need a certain amount of oxygen for breathing. Oxygen is released into the atmosphere by forest vegetation (about 1.3 t of oxygen per ton of woody mass, in a day) through its chlorophyll assimilation process. A mature tree produces about 1.7 kg of oxygen/day, which means the oxygen requirement of a human for three days [32]. Also, in the same process of tree photosynthesis, a large amount of carbon dioxide is consumed, approximatively 1.8 t per one cubic meter of wood. On hot days, a hectare of a highly productive forest consumes an average of 220–280 kg of carbon dioxide per day. Wood production of the three forestry species in this study is different: linden at 350 $\text{m}^3$/ha, balsa at 550 $\text{m}^3$/ha, and paulownia at 950 $\text{m}^3$/ha. The volume of timber obtained after the first 3–4 years of paulownia is 0.3–0.8 $\text{m}^3$/tree, i.e., 200–500 $\text{m}^3$/ha. The medium age of exploitation is 75 years for linden, 50 years for balsa, and 15 years for paulownia [33,34]. To establish the ecological factor for all three species, the unitary coefficient for linden forests was considered, which was used as a reference for comparison with the other two forest species (paulownia and balsa).

**The comparative analysis of the three species.** The method of comparative analysis of the 3 wood species took linden wood as the element of comparison because it is currently considered the most homogeneous species, especially from the point of view of the ratio of tangential shrinkage to radial shrinkage. Several analysis criteria were used, namely anatomical structure, density, resistance, color, shrinkage and swelling, and the ratio of tangential shrinkage to radial shrinkage. All the analyzed characteristics for paulownia and balsa woods were compared to those of linden to obtain positive/negative percentages, using the following relationship (Equation (10)):

$$R = \frac{C_{b,p} - C_l}{C_l} \cdot 100 \ [\%] \tag{10}$$

where $C_{b,p}$ is the characteristic of balsa/paulownia wood, and $C_l$ is the characteristic of linden wood.

Based on the all obtained values of characteristics, it was cumulatively determined which of the two species (balsa and paulownia) is more appropriate as a possible replacement for linden wood in its uses.

**Statistical analysis.** Standard deviation and arithmetic mean were calculated for each group of tests. Also, the regression tendences and the Pearson coefficients $R^2$ were

identified on the graphs obtained using Microsoft Excel 2019 (Microsoft Corp., Redmond, WA, USA). With the help of the statistical program Minitab 18 (Penn State University, State College, PA, USA) and its generated graphs, the mean, standard deviation, *p*-value, and Anderson–Darling coefficient of all tested values were found, with a confidence interval of 95% or an error of 0.05.

### 3. Results

*3.1. Physical and Mechanical Wood Properties (Base on the Data from Reference [1])*

Table 1 shows a comparison between linden, balsa, and paulownia species, with the first two species being often used in restoration, consolidation, and conservation processes [12], and the last one being a possible species to be used in this field.

**Table 1.** Comparison between the wood species of linden, balsa, and paulownia [1].

| No. | Wood Species | | Linden | Balsa | Paulownia |
|---|---|---|---|---|---|
| 1. | Distribution | | Europe | America | Asia and North America |
| 2. | Tree size—max height and diameter, in m | | 20–40; 1.5–2 | 18–28; 1–1.2 m | 10–20 m; 0.6–1.2 |
| 3. | Density kg/m³ | 0% | 420 | 120 | 250 |
| 4. | | 12% | 535 | 150 | 280 |
| 5. | MOR, MPa | | 85.4 | 19.6 | 37.8 |
| 6. | MOE, GPa | | 11.71 | 3.71 | 4.38 |
| 7. | Compressive strength, MPa | | 44.8 | 11.6 | 20.7 |
| 8. | Shrinkage, % | Radial | 5.0 | 2.3 | 2.5 |
| 9. | | Tangential | 7.5 | 6.2 | 3.9 |
| 10. | | Volumetric | 12.6 | 8.6 | 6.5 |
| 11. | | T/R ratio | 1.5 | 2.6 | 1.6 |
| 12. | Quality coefficient, MOR/$\rho$ | | 159.6 | 130.6 | 135.0 |
| 13. | Quality coefficient, $\sigma_c/\rho$ | | 83.7 | 77.3 | 73.9 |

Even if the three species have different distribution areas (linden in Europe, balsa in America, and paulownia in Asia and North America), they are found together in all corners of the world. Linden is considered a uniform species, even if it has a high density, while balsa has a density reduction of 71.5%, and paulownia has a density reduction of 47.6% (for densities at 12% moisture content) (Table 1, Figures 5 and 6). Thus, from the point of view of density, paulownia is closer to linden than balsa.

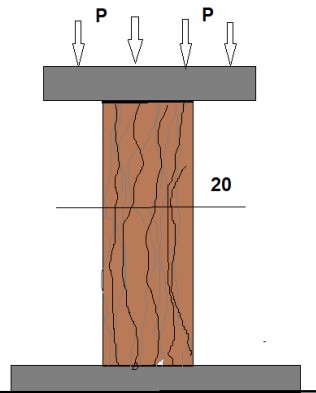

**Figure 5.** Compressive strength of a wood specimen.

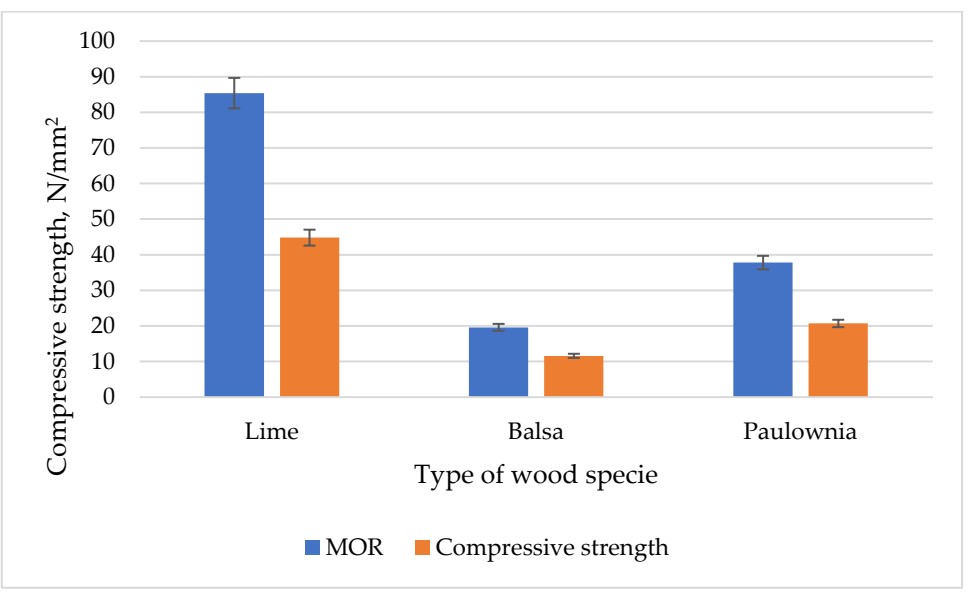

**Figure 6.** Resistance to compression of lime, balsa, and paulownia woods.

From the point of view of resistance to static bending (MOR), linden wood had the highest value due to its maximum density, with the decrease in resistance being 77% in the case of balsa wood and 55.7% in the case of paulownia wood. These decreases in bending strength (MOR) were greater than those in density, meaning that the dependence relationship was not proportional. From the point of view of the modulus of elasticity, linden wood was the most elastic, with the decrease in elasticity being 68.3% in the case of balsa wood and 62.5% in the case of paulownia wood. From the point of view of compressive strength, linden wood had the highest resistance at 44.8 N/mm$^2$, with the strength of balsa wood decreasing by 74.1%, and that of paulownia wood by 53.7%. Thus, from the point of view of compression resistance, paulownia wood is very close to that of linden, especially after its lower density has been taken into account.

Following the analysis shown in Table 1, from the point of view of shrinkage, linden wood had the highest values with values of 5% in the radial direction, 7.5% in the tangential direction, and 12.6% in volume, and the lowest values were identified for paulownia wood with values of 2.5% in the radial direction, 3.9% in the tangential direction, and 6.5% in volume. There were very large decreases in the shrinkage of paulownia wood (compared to that of linden wood) in the radial direction by 50%, in the tangential direction by 48%, and in volume by 48.4%. The best factor of homogeneity in relation to humidity and water is the ratio between tangential and radial shrinkage; this ratio was very good in the case of linden and paulownia wood, but it was also above the average value of 2.0 for all species, in the case of balsa wood (Figures 7 and 8).

Figure 8 and Table 2 show that the density of each wood species is different depending on its moisture content, taking as the reference values the moisture content values of absolutely dry wood (MC = 0%), at 12%, and above the saturation point of fiber (MC > 30%). The first two values (at 0% and 12% moisture content) are common ones, but the third value (over the fiber saturation point) is also important because, when the value is above the saturation point of fiber, wood no longer swells and shrinks (it does not change its dimensions, with the volume remaining constant), but it still absorbs water (increases its mass), thus significantly increasing the density. It was observed that, although the immersion time was the same, the moisture content increased more in balsa wood, which is a less dense species (with more free holes for water to penetrate).

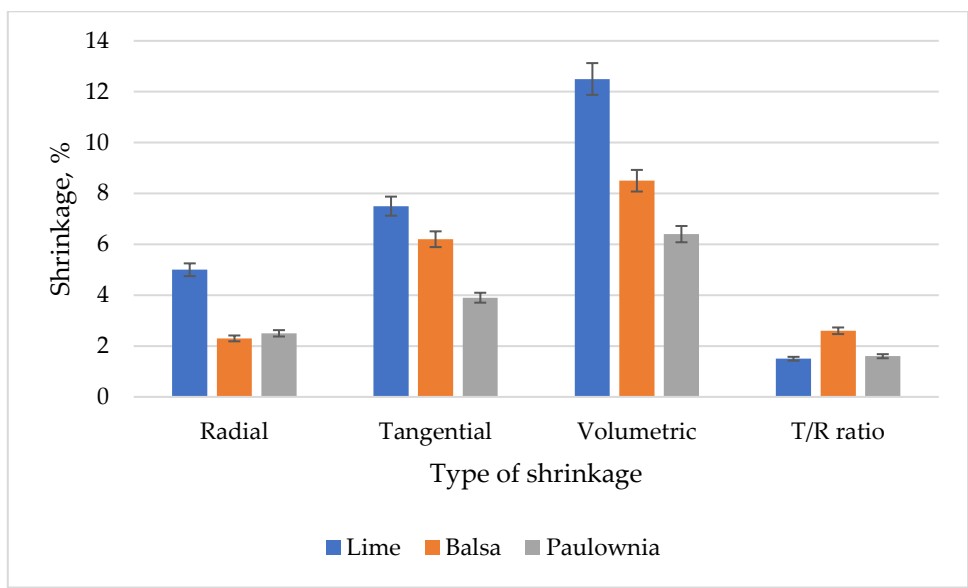

**Figure 7.** Shrinkage of lime, balsa, and paulownia woods.

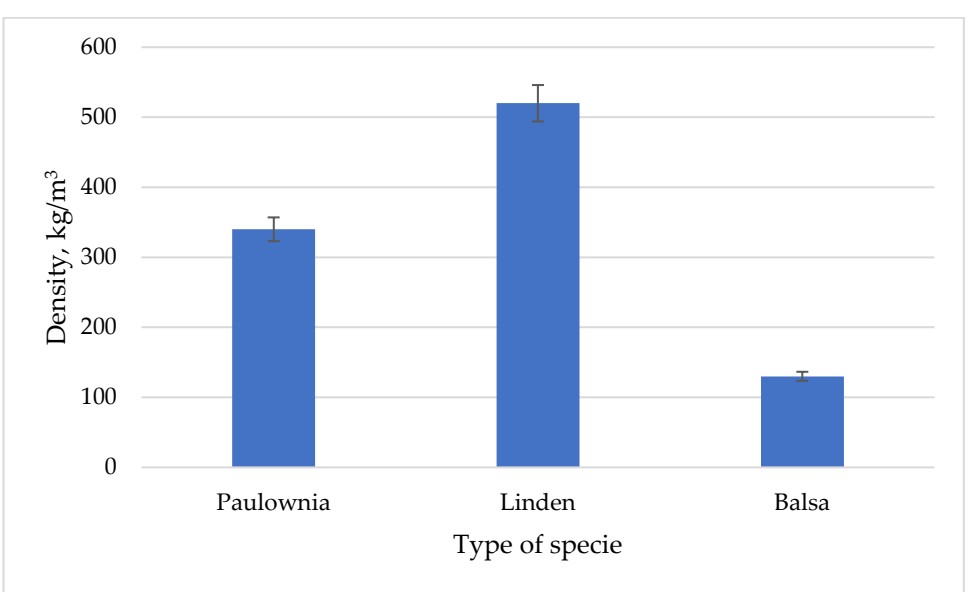

**Figure 8.** Density of the three wood species.

**Table 2.** Results regarding the density of the three wood species.

| Species | Density, kg/m³, Related to Moisture Content (MC) | | |
| --- | --- | --- | --- |
| | MC = 0% | MC = 12 % | Over the Fiber Saturation Point |
| Paulownia | 310 | 340 | 380, MC = 30% |
| Linden | 460 | 520 | 580, MC = 40% |
| Balsa | 120 | 130 | 210, MC = 85% |

### 3.2. Experimental Results for Wood Density

For comparison, the density of the samples in an absolutely dry state was examined (Figure 9). It was observed that the density of linden wood was the highest (461 kg/m³), and that of balsa wood was the lowest (118 kg/m³).

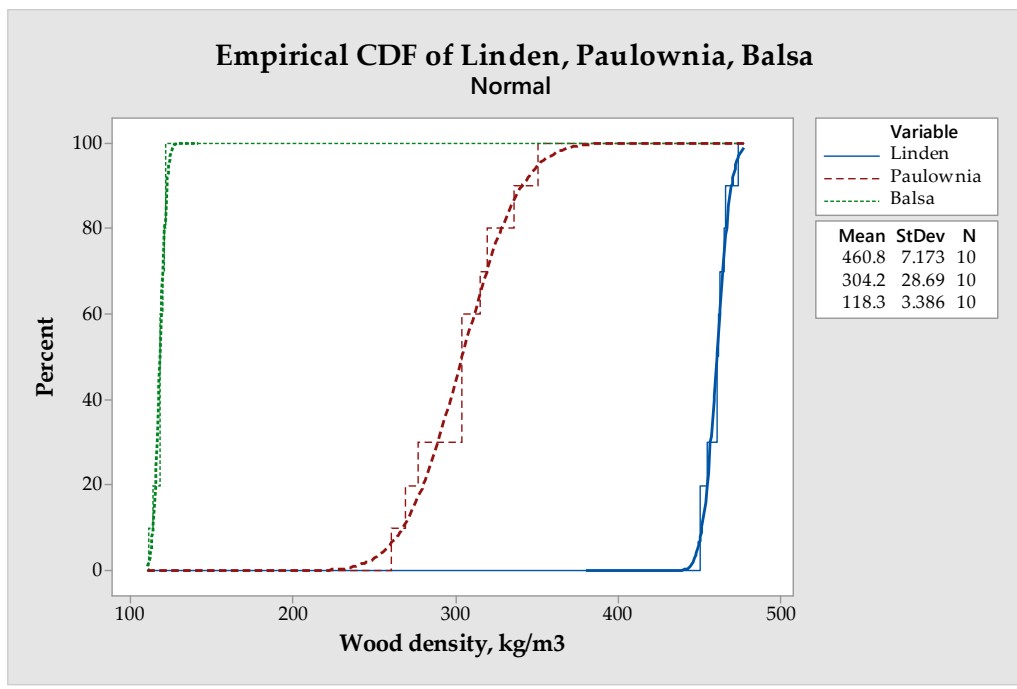

**Figure 9.** Empirical cumulative distribution function (CDF) of wood density.

Bearing in mind that density is greatly influenced by moisture content, to find the density values depending on the moisture content, the following relationship was used (Equation (11)):

$$\rho_{MC} = \rho_0 \cdot \frac{1 + MC}{1 + \rho_0 \cdot MC} \tag{11}$$

where $\rho_0$ is the oven-dried wood density, in g/cm$^3$, and *MC* is the moisture content, in decimal.

Also, to find the maximum moisture content ($MC_{max}$) of each wood species, the following relationship was used (Equation (12)):

$$MC_{max} = \frac{1.5 - \rho_0}{1.5 \cdot \rho_0} \tag{12}$$

The obtained value of wood densities are presented in Table 3.

**Table 3.** Density of wood species at different moisture content.

| Wood Species | Density at Different Moisture Content, $\rho_u$, kg/m$^3$ | | | | | | | |
|---|---|---|---|---|---|---|---|---|
| | 0% | 12% | 30% | 60% | 90% | 120% | 150% | $MC_{max}$ |
| Linden | 461 | 489 | 526 | 578 | 619 | 672 | 681 | 681 ($MC_{max}$ = 150%) |
| Paulownia | 304 | 328 | 362 | 411 | 454 | 490 | 522 | 612 ($MC_{max}$ = 262%) |
| Balsa | 118 | 130 | 148 | 176 | 202 | 227 | 250 | 540 ($MC_{max}$ = 780%) |

### 3.3. Colorimetry of the Three Wood Species Using CIEL*a*b* Space

#### 3.3.1. The Color of Paulownia Wood before and after Planning

Immediately after cutting the log into lumber, the paulownia wood darkened in color to a light gray. This coloring was observed on a small thickness of about 0.5–0.7 mm, due to the oxidation of wood sap when it came into contact with atmospheric air. This color was kept as long as the wood was not processed via planning. The dark color disappeared when the timber was planned to a thickness of at least 0.7 mm, obtaining the real beautiful color of paulownia wood surface (Table 4).

**Table 4.** The color changes of paulownia timber wood before and after planning.

| No. | Timber Surface before Planning | | | Timber Surface after Planning | | |
|---|---|---|---|---|---|---|
| | L* | a* | b* | L* | a* | b* |
| 1. | 4.725 | −3.36 | −12.86 | 63.92 | −23.41 | 21.13 |
| 2. | 2.435 | −15.03 | −10.16 | 64.49 | −23.36 | 21.18 |
| 3. | 3.293 | −1.45 | −6.68 | 64.29 | −23.66 | 21.17 |
| 4. | 5.779 | −7.65 | −10.47 | 63.4 | −22.87 | 20.41 |
| 5. | 8.369 | −19.41 | −11.78 | 55.52 | −22.94 | 20.61 |
| 6. | 7.734 | −5.66 | −7.43 | 58.43 | −22.28 | 20.89 |
| 7. | 6.451 | −12.71 | −9.87 | 61.72 | −21.89 | 21.75 |
| 8. | 3.478 | −6.59 | −12.66 | 59.39 | −24.86 | 22.02 |
| 9. | 4.673 | −2.99 | −8.64 | 64.15 | −22.89 | 21.17 |
| 10. | 7.184 | −13.67 | −9.17 | 62.63 | −21.64 | 20.97 |
| Mean | 5.412 | −8.852 | −9.97 | 61.794 | −22.98 | 21.13 |
| SD | 1.913 | −0.569 | 0.197 | 2.886 | −0.884 | 0.453 |

Figure 10 shows the degree of black and white of the paulownia timber before and after planning, clearly demonstrating an increase in color from white to black (from 5.41, to 61.79 units).

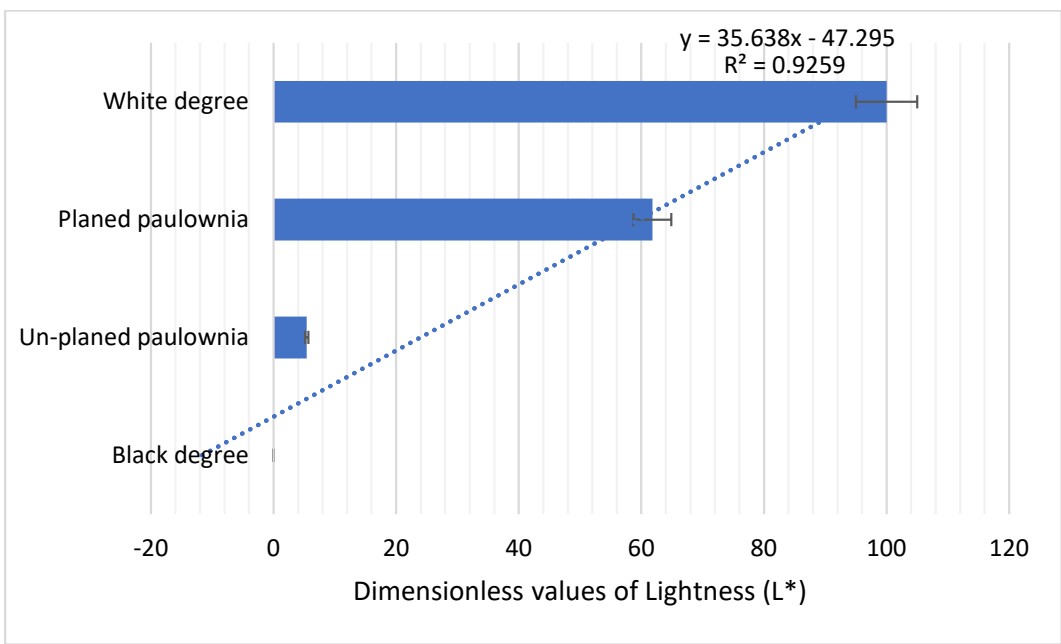

**Figure 10.** The degree of white/black (L*) of paulownia timber before and after planning.

Figures 11 and 12 show the green–red and blue–yellow degrees of paulownia timber before and after planning. There was a decrease in the degree of green–red from −8.8 to −22.9 units (14.1 units) and a large increase in the degree of blue–yellow from −9.9 to 21.1 units (an increase of 31 units).

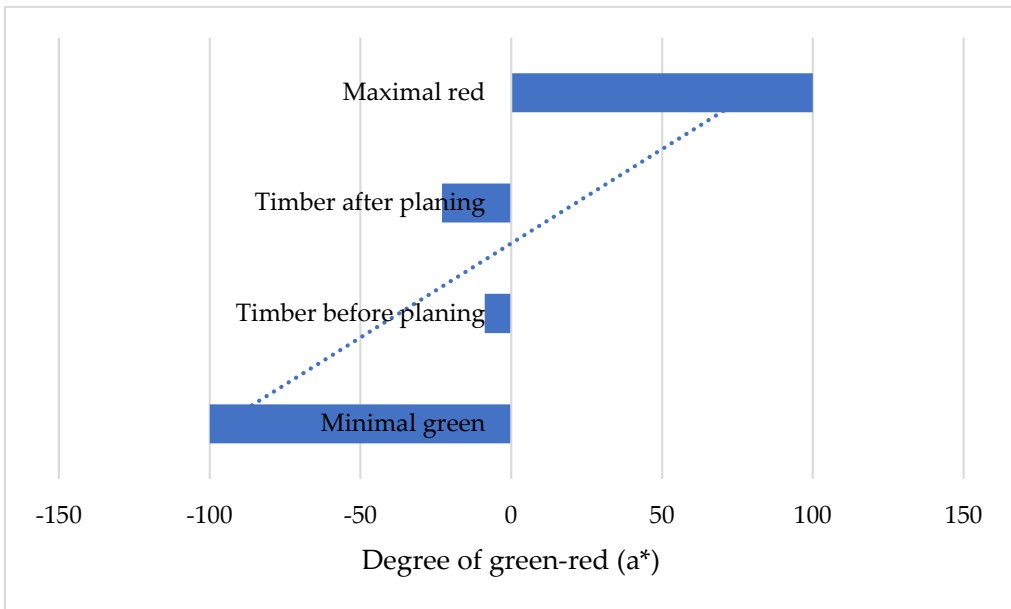

**Figure 11.** Degree of green–red (a*) of paulownia timber before and after planning.

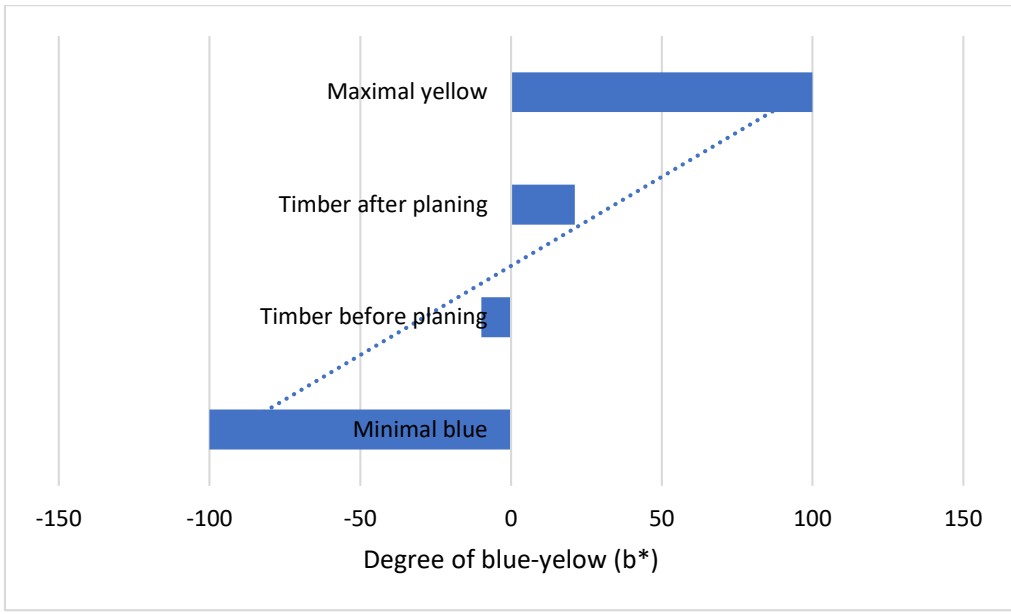

**Figure 12.** Degree of yellow–blue (b*) of paulownia timber before and after planning.

### 3.3.2. Results of the Color Analysis of the Three Species

The comparative color analysis of the three species is highlighted in Figure 13, indicating a higher luminance for balsa (Figure 10), a degree of red that is accentuated negatively (35.28–37.62) for all analyzed wood species, and a very low degree of yellowness (3.04–3.66).

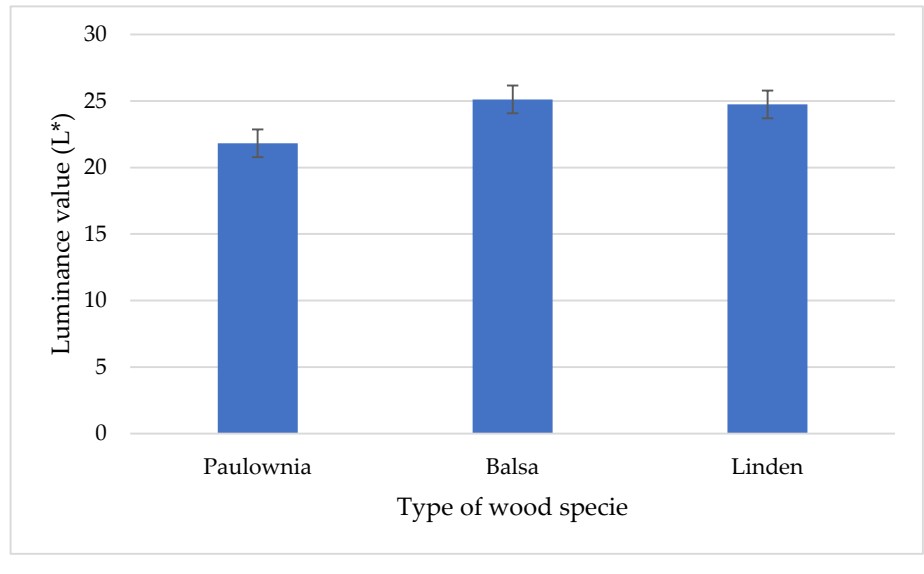

(**a**)

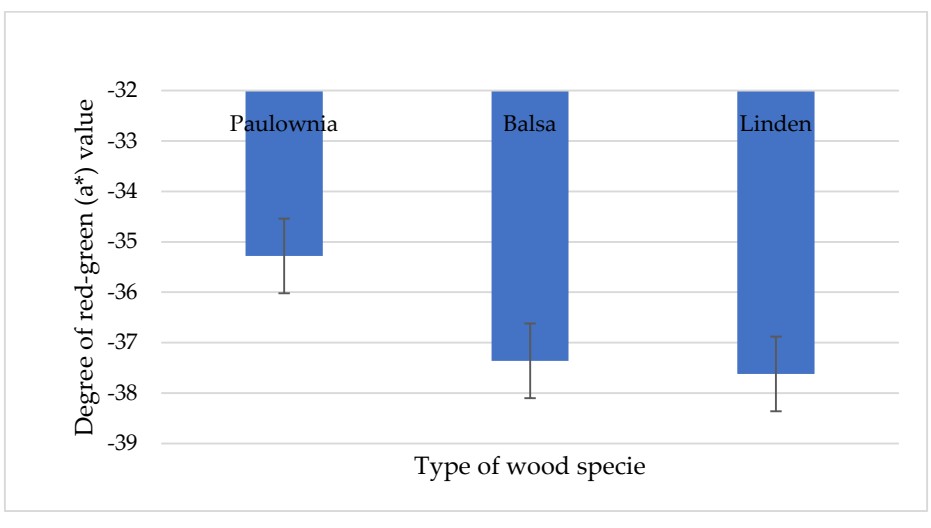

(**b**)

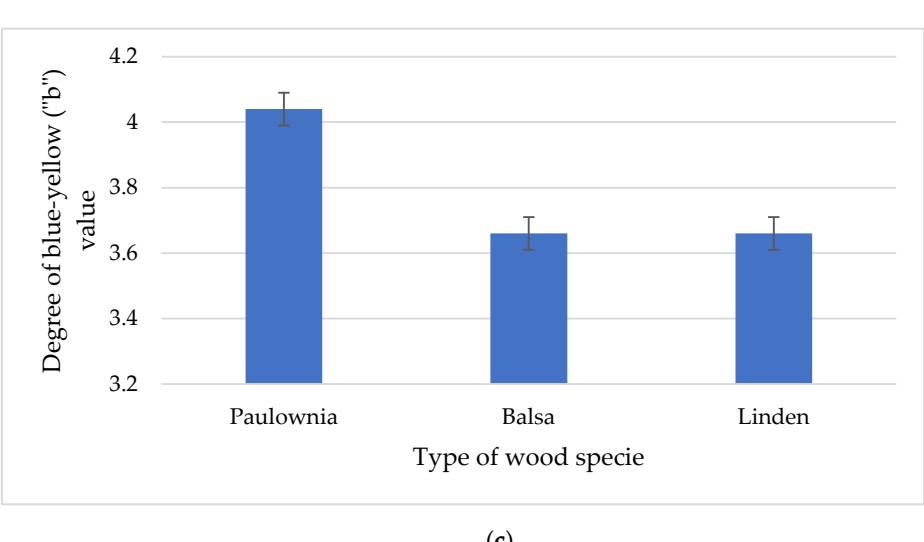

(**c**)

**Figure 13.** Values of the CIELab color space for luminance (L*) (**a**), for the degree of red–green (a*) (**b**), and for the degree of blue–yellow (b*) (**c**).

It could be observed that both luminance (L*), red–green contrast (a*), and blue–yellow contrast (b*) had some very close values, which means that from a coloristic point of view, the three analyzed species are appropriate and homogeneous (Figure 13). Furthermore, when the "a*" and "b*" values of the three species were plotted in a two-dimensional plane (Figure 14), it could be noticed that all three points are located in the fourth quadrant of the plane, and the points are very close to each other. Thus, from a color point of view, the three species are very similar.

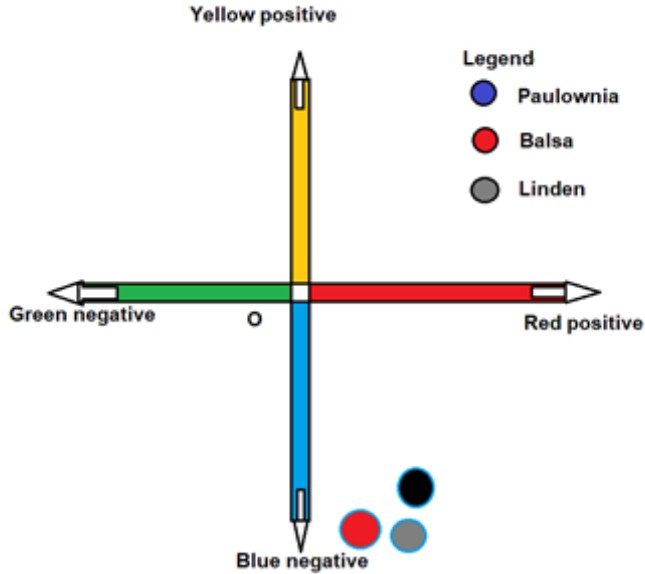

**Figure 14.** Positioning the color points of linden, balsa, and paulownia wood in a two-dimensional plane (positive–negative).

### 3.4. Results for Water Absorption

The results for water absorption of the three wood species are presented in Table 5 and Figure 15. It is clearly demonstrated that paulownia wood had the lowest water absorption (27.3%) and balsa wood had the highest one (85.3%).

**Table 5.** Results for water absorption of the three wood species.

| No. | Paulownia Mass, g | | WA, % | Linden Mass, g | | WA, % | Balsa Mass, g | | WA, % |
|---|---|---|---|---|---|---|---|---|---|
| | Final | Initial | | Final | Initial | | Final | Initial | |
| 1 | 6.31 | 4.98 | 26.70 | 8.61 | 6.11 | 40.91 | 2.86 | 1.53 | 86.92 |
| 2 | 5.51 | 4.28 | 28.73 | 8.66 | 6.26 | 38.33 | 2.93 | 1.58 | 85.44 |
| 3 | 6.1 | 4.82 | 26.55 | 8.84 | 6.43 | 37.48 | 2.71 | 1.5 | 80.66 |
| 4 | 6.46 | 5.07 | 27.41 | 8.78 | 6.42 | 36.76 | 2.91 | 1.58 | 84.17 |
| 5 | 6.56 | 5.15 | 27.37 | 8.41 | 6.1 | 37.86 | 2.82 | 1.53 | 84.31 |
| 6 | 6.41 | 5.05 | 26.93 | 8.5 | 6.13 | 38.66 | 2.89 | 1.51 | 91.39 |
| 7 | 6.6 | 5.18 | 27.41 | 8.83 | 6.34 | 39.27 | 2.62 | 1.39 | 88.48 |
| Mean | 6.279 | 4.932 | 27.305 | 8.66 | 6.26 | 38.47 | 2.82 | 1.52 | 85.92 |
| St. Dev | 0.38 | 0.311 | 0.723 | 0.17 | 0.14 | 1.35 | 0.114 | 0.064 | 3.440 |

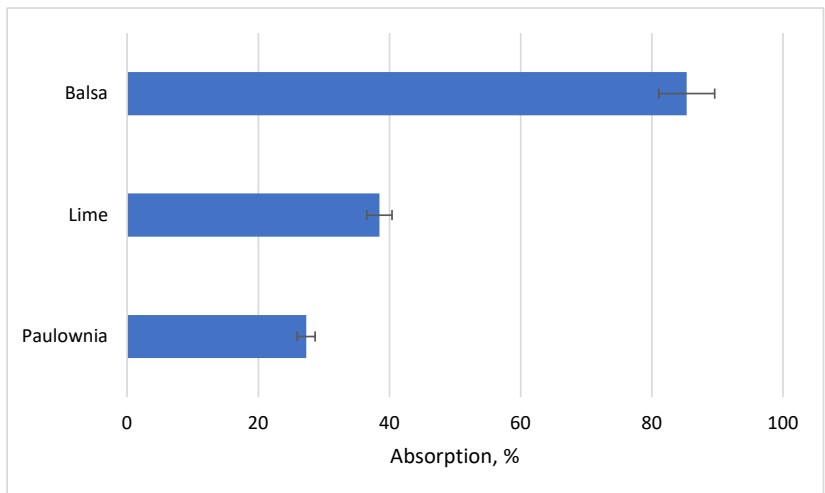

**Figure 15.** Water absorption of the three wood species.

### 3.5. Results for Wood Shrinkage

The results for wood shrinkage are presented in Table 6 and Figure 16. It can be seen that the largest volumetric shrinkage was found in linden wood (8.29%), and a smaller volumetric shrinkage was found in paulownia and balsa wood (4.13% and 3.39%, respectively).

**Table 6.** Wood shrinkage for paulownia, linden, and balsa.

| No. | Paulownia, mm | | | | | Linden, mm | | | | | Balsa, mm | | | | |
|---|---|---|---|---|---|---|---|---|---|---|---|---|---|---|---|
| | T | R | L | V | T/R | T | R | L | V | T/R | R | T | L | V | T/R |
| 1 | 2.50 | 1.61 | 0.034 | 4.15 | 1.55 | 4.30 | 3.37 | 0.14 | 7.81 | 1.27 | 1.08 | 1.53 | 0.40 | 3.01 | 1.41 |
| 2 | 1.64 | 0.88 | 0.034 | 2.55 | 1.86 | 4.46 | 3.18 | 4.13 | 11.7 | 1.40 | 1.84 | 1.95 | 0.48 | 4.27 | 1.05 |
| 3 | 2.57 | 1.31 | 0.209 | 4.09 | 1.96 | 3.81 | 3.69 | 0.00 | 7.50 | 1.03 | 1.61 | 1.64 | 0.27 | 3.52 | 1.01 |
| 4 | 2.78 | 0.87 | 0.138 | 3.78 | 3.19 | 4.54 | 3.45 | 0.28 | 8.27 | 1.31 | 1.55 | 1.66 | 0.20 | 3.42 | 1.07 |
| 5 | 2.87 | 1.44 | 0.172 | 4.48 | 1.99 | 4.82 | 3.05 | 0.17 | 8.04 | 1.58 | 0.66 | 1.23 | 0.27 | 2.16 | 1.86 |
| 6 | 2.74 | 1.69 | 0.550 | 4.98 | 1.62 | 4.97 | 2.21 | 0.49 | 7.67 | 2.24 | 1.58 | 1.86 | 0.17 | 3.60 | 1.17 |
| 7 | 2.81 | 1.70 | 0.345 | 4.86 | 1.65 | 3.99 | 2.80 | 0.21 | 6.99 | 1.42 | 1.57 | 1.86 | 0.34 | 3.77 | 1.18 |
| M | 2.56 | 1.36 | 0.212 | 4.13 | 1.88 | 4.41 | 3.11 | 0.77 | 8.29 | 1.4 | 1.42 | 1.67 | 0.31 | 3.39 | 1.25 |
| ST | 0.43 | 0.36 | 0.184 | 0.81 | 0.52 | 0.42 | 0.49 | 1.49 | 1.59 | 0.38 | 0.4 | 0.24 | 0.1 | 0.66 | 0.16 |

T—tangential; R—radial; L—longitudinal; M—mean; ST—standard deviation.

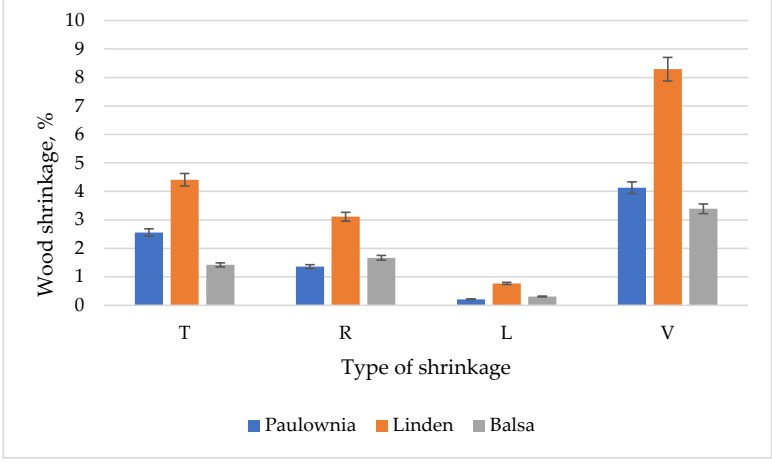

**Figure 16.** Shrinkage of the wood species. T—tangential; R—radial; L—longitudinal; V—volumetric.

### 3.6. Brinell Hardness

The Brinell hardness (BH) results are presented in Figure 17, which highlights the maximum value for linden wood (20.4 N/mm$^2$) and the minimum value for balsa wood (11.1 N/mm$^2$). It is also noticeable the value of Brinell hardness is higher in the radial section for all three analyzed species.

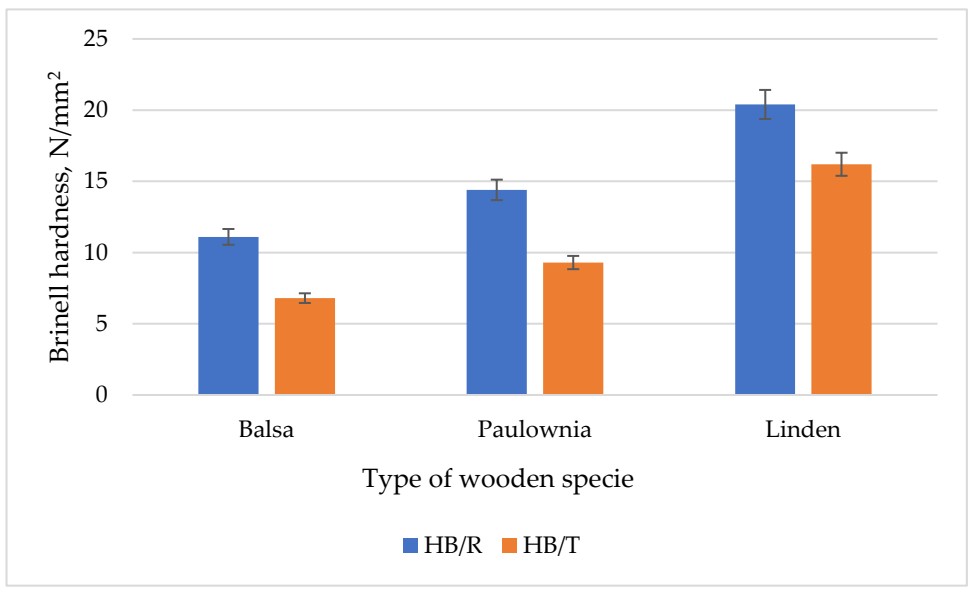

**Figure 17.** Brinell hardness for balsa, paulownia, and linden wood in the radial (HB/R) and tangential (HB/T) sections.

### 3.7. Results for Compressive Strength

The results regarding compressive strength are presented in Figure 18, where it can be seen that the maximum compression resistance was obtained for linden wood (52.9 N/mm$^2$), and the minimum value was obtained for balsa wood (9.1 N/mm$^2$).

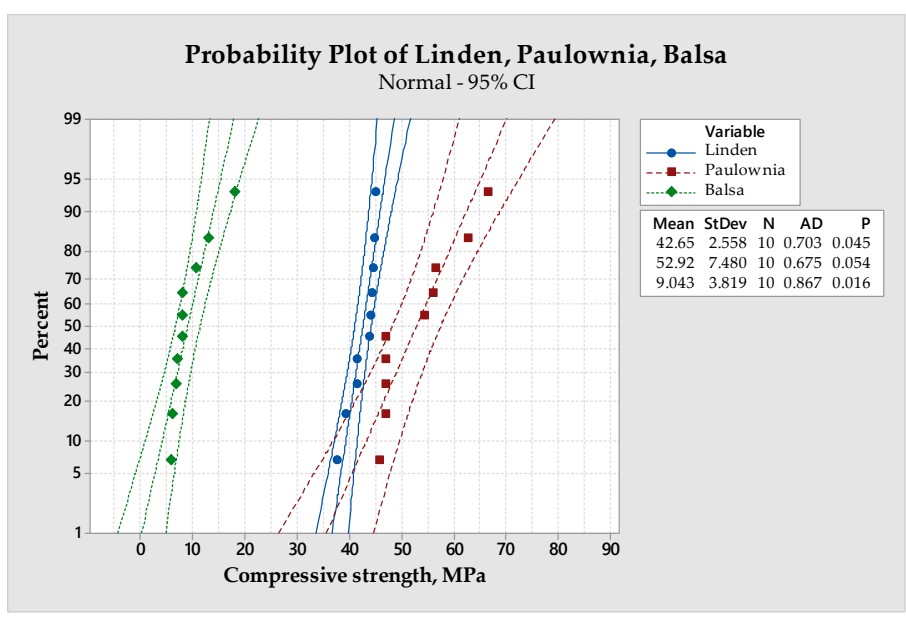

**Figure 18.** The compressive strength of all three wood species.

### 3.8. Results for the Modulus of Resistance (MOR) to Static Bending

The results obtained for the modulus of resistance (MOR) upon static bending for the three wood species corresponded with their density, showing a value of 92.3 N/mm$^2$ in the case of linden wood, 62.1 N/mm$^2$ in the case of paulownia wood, and 14.1 N/mm$^2$ in the case of balsa wood (Figure 19). The influence of wood density on the modulus of resistance (MOR) is also demonstrated in Figure 20, with paulownia wood as an example.

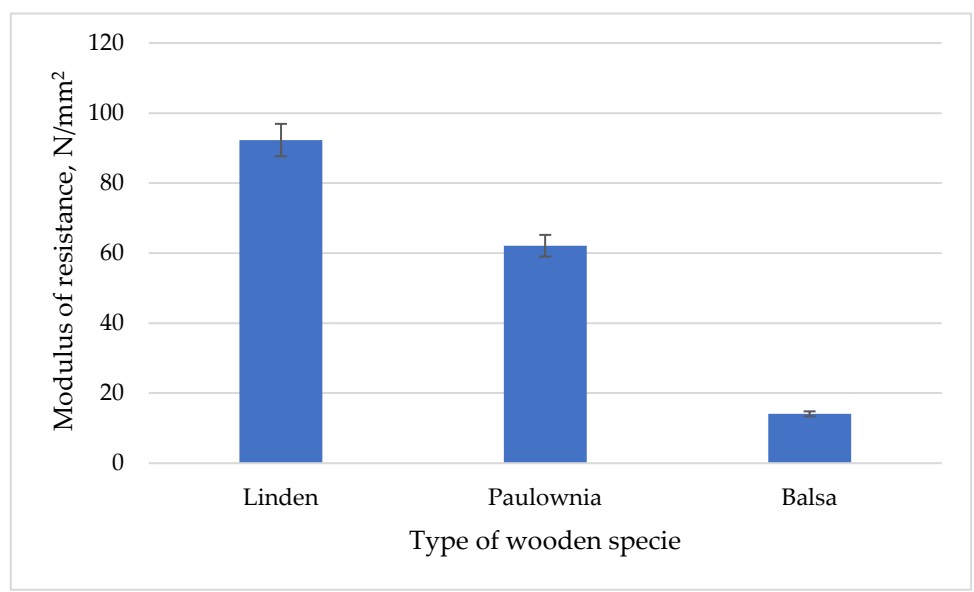

**Figure 19.** Modulus of resistance (MOR) obtained for the three wood species.

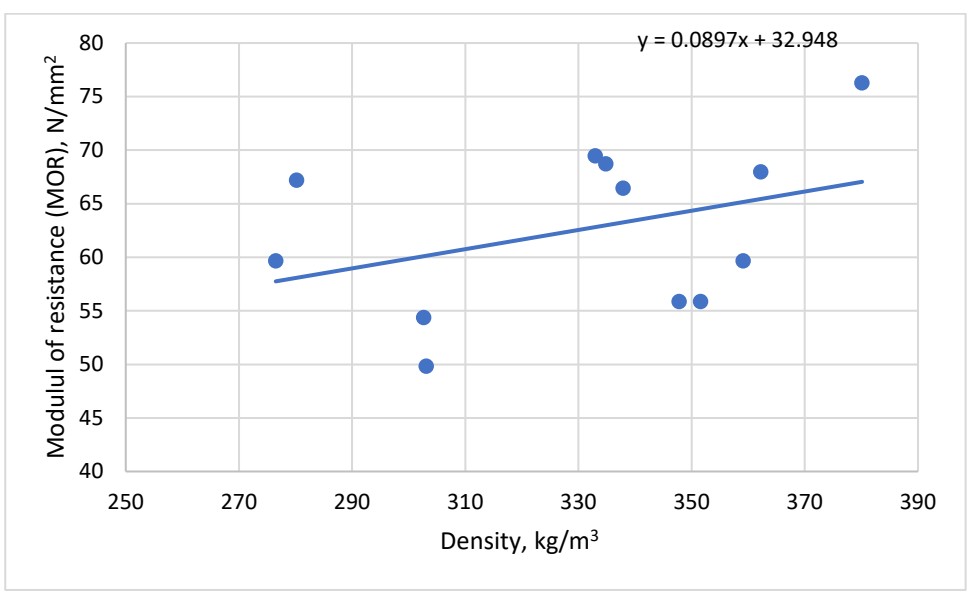

**Figure 20.** Influence of density on the modulus of resistance (MOR) to static bending strength for paulownia.

### 3.9. Results for the Modulus of Elasticity (MOE) to Static Bending

The results of the modulus of elasticity (MOE) to static bending are presented in Figure 21, where it is demonstrated (as in the case of the modulus of resistance) that the maximum value was obtained for linden wood (6782 N/mm$^2$), and the minimum value was obtained for balsa wood (927 N/mm$^2$). A direct proportionality was observed between wood density and the modulus of elasticity (MOE) for paulownia wood (because the graphs

of the influence of density on the modulus of elasticity for linden and balsa were similar) (Figure 22).

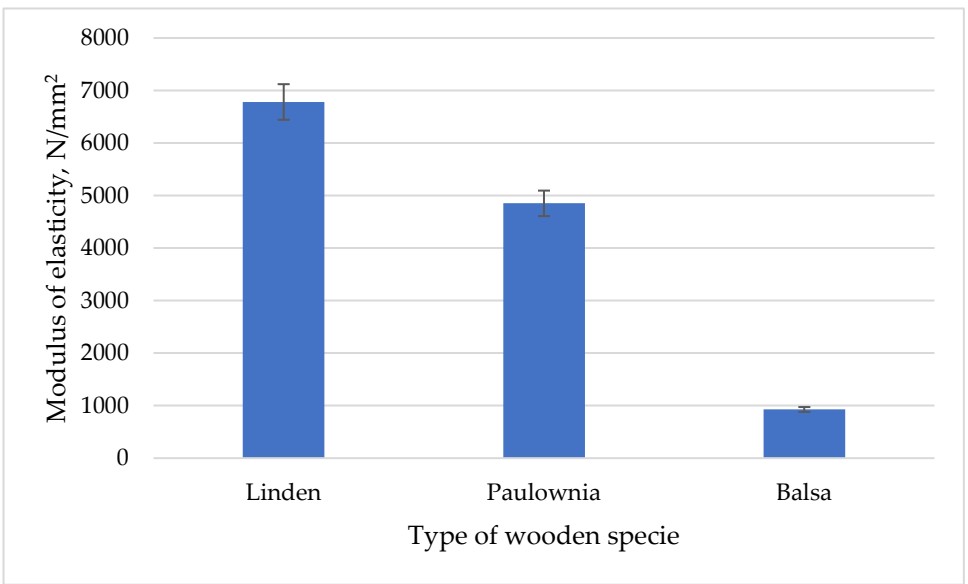

**Figure 21.** Modulus of elasticity obtained for the three wood species.

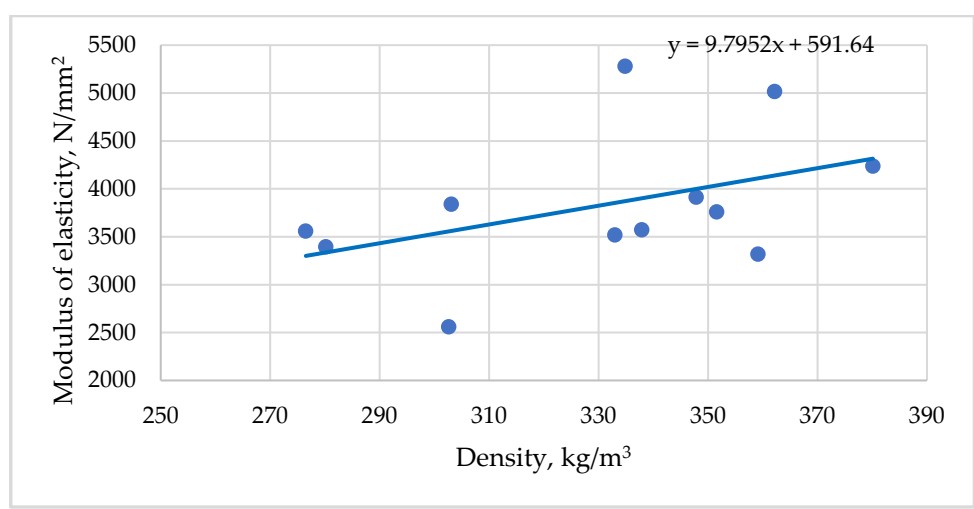

**Figure 22.** Influence of density on the modulus of elasticity (MOE) for paulownia wood.

*3.10. Ecological Influences*

The results related to the ecology of the three forest species started from the absorption of carbon dioxide of 1.8 $t/t$ wood and removal of oxygen of 1.3 $t/t$ wood, but also considering the different growth rates of the three species, which are 350 m$^3$/ha in the case of linden tree, 550 m$^3$/ha in the case of balsa tree, and 850 m$^3$/ha in the case of paulownia tree. Taking into account the densities of these wood species, a quantity of mass of 650 t/ha was obtained in the case of linden, 3600 t/ha in the case of balsa, and 3400 t/ha in the case of paulownia. In this way, 845 t of $O_2$/t wood and 1170 t of $CO_2$/ha in the case of linden, 4680 t of $O_2$/wood and 6480 t $CO_2$/ha in the case of balsa, and 4420 t $O_2$/t wood and 6120 t $CO_2$/ha in the case of paulownia were obtained. If the ecological factor was a unitary one in the case of linden forests, then a factor of 5.5 was obtained for balsa forests and a factor of 5.2 was obtained for paulownia forests.

## 4. Discussion

To be able to have a conclusive discussion regarding the results obtained in this work, all results are grouped in Table 7. Also, this table includes the comparison values obtained for paulownia and balsa wood to those obtained for linden wood, corresponding to one of the relationships (Equation (10)).

**Table 7.** Comparison of the characteristics of the balsa and paulownia wood species related to linden one.

| No. | Characteristics | Linden | Paulownia | | Balsa | |
|---|---|---|---|---|---|---|
| | | Value | Value | Eq. 11% | Value | Eq. 11% |
| 1. | Unit density, $\rho_0$, kg/m$^3$ | 461 | 304 | −34.0 | 118 | −74.4 |
| 2. | Luminance | 21.8 | 25.1 | +15.1 | 24.7 | +13.3 |
| 3. | a* value | −35.2 | −37.3 | −5.9 | −37.6 | −6.8 |
| 4. | b* value | 4.0 | 3.6 | −10.0 | 3.6 | −10.0 |
| 5. | Absorption, % | 38 | 27 | +7.8 | 85 | −123.6 |
| 6. | Tangential shrinkage (TS), % | 4.4 | 2.5 | +43.1 | 1.4 | +68.1 |
| 7. | Radial shrinkage (RS), % | 3.1 | 1.3 | +58.1 | 1.6 | +48.3 |
| 8. | Ratio, TS/RS | 1.4 | 1.8 | −28.5 | 1.2 | +14.2 |
| 9. | Compressive strength, N/mm$^2$ | 52.9 | 42.5 | −2.3 | 9.1 | −82.7 |
| 10. | Modulus of resistance, N/mm$^2$ | 92.3 | 62.1 | −32.7 | 14.1 | −84.7 |
| 11. | Modulus of elasticity, N/mm$^2$ | 6782 | 4852 | −28.4 | 927 | −86.3 |
| 12. | Brinell hardness (R), N/mm$^2$ | 20.4 | 14.4 | −29.4 | 11.1 | −45.8 |
| 13. | Ecological coefficient $O_2$; $CO_2$ | 1 | 5.5 | +550 | 5.2 | +520 |
| 14. | Quality coefficient, MOR/$\rho_{0(g/cm^3)}$ | 200.2 | 204.2 | +1.9 | 119.4 | +0.8 |
| 15. | Quality coefficient, $\sigma_c/\rho_{0(g/cm^3)}$ | 114.7 | 139.8 | +21.8 | 77.1 | −32.7 |
| | Total points | ---- | ---- | +526.6 | ----- | +117.7 |

The first analysis of the results shown in Table 7 was a comparison between the characteristics of the three species, namely how close the characteristics of paulownia and balsa are to those of linden wood. It was clearly observed that paulownia wood was much more appropriate as a replacement for linden wood with a total of 526.6 points, thus being 4.4 times more appropriate than balsa wood.

Paulownia wood had a density lower than that of linden by 34%, but higher than that of balsa wood by 1.62 times. It was observed that the experimental value of balsa wood density was closest to that obtained by other researchers [1], with the difference only 1.6% lower. In the case of linden wood, the difference between the two values was higher than 9.7%, and in the case of paulownia wood, the difference was higher than 21.6%. From the point of view of density, the three analyzed species are different as the differences are greater than 5–10% between them, paulownia wood being more appropriate as a replacement for linden [35,36]. Some authors [17] found a wood density of about 300 kg/m$^3$ for paulownia, which was very close to the value of 304 kg/m$^3$ of this research.

The analysis of color parameters with regard to the natural color of planed wood showed that from a color point of view, the three wood species are very appropriate (L = 21.8–25.1; a = −(35.2–37.6); and b = 3.6–4), with the points of all three species being positioned in the fourth trigonometric quadrant. Appropriate values were also found by other authors [5,18], and a similar value of paulownia was obtained by others [17].

The water absorption of these three wood species was dependent on their porosity; it increased with a decrease in the density of these species. The exception to this rule was paulownia wood, which had a lower absorption than linden wood. The explanation for this phenomenon could be attributed to the microscopic structure of this wood species

(Figure 1) and the fact that it has pores filled with cellulosic formations that prevent the penetration of water into the wood.

As a general trend, the wood shrinkage of these three wood species decreased with a decrease in density (regardless of whether the direction of analysis was radial or tangential), with balsa wood having the lowest shrinkage (1.4% in the tangential direction and 1.6% in the radial direction) [37]. The values of the ratio of tangential shrinkage to radial shrinkage showed that these three wood species are uniform, with balsa wood and linden wood having the lowest shrinkage coefficients.

The Brinell hardness of the three wood species had values corresponding to the values reported in other research studies in the field [38,39]), but there were also small differences between them. For example, the Brinell hardness of balsa wood was only 22.8% lower than that of paulownia wood, although the density difference between the two wood species was 74.4%.

The compressive strength of these three species (Figure 17) was different and depended on density, with a higher compressive resistance (52.9 N/mm$^2$) being obtained for linden, which had a higher wood density (461 kg/m$^3$). As a general trend, from the point of view of compression strength, paulownia wood has a value closer to linden wood. The values obtained in this research were slightly higher than those reported in the literature [1,40], being higher by 15.3% for linden, 8.4% for paulownia, and 8.2% for balsa wood.

The bending strength (MOR) was in accordance with the density of the wood species, with higher densities corresponding to higher resistances. The dependence relationship between the densities and resistances of these three species (Figure 19) was approximated as linear; from this point of view, paulownia wood has superior characteristics compared to the other two species. Differences between the values obtained in this research and those reported in the literature [1,41] were also identified; in this case the difference was about 7.4% for linden wood. The modulus of elasticity (MOE) maintained the same trend as the modulus of resistance (MOR), showing a decrease along with a decrease in density. However, the differences between the values of the three species have increased, as the modulus of elasticity of paulownia wood (4852 N/mm$^2$) becoming closer to that of linden wood (6782 N/mm$^2$), while the modulus of elasticity of balsa wood (927 N/mm$^2$) moving further away. The values obtained in this work differed slightly from those reported in the literature [1]. For instance, there was a large difference of about 4.72 times in the case of balsa wood. In the direction of what was mentioned above, in the reference [23], an MOR value of 111 N/mm$^2$ and an MOE value of 11,479 N/mm$^2$ were found for beech wood, but with a much higher density than the species analyzed in this study, of 727 kg/m$^3$.

Based on the results shown in Table 7, due to the different rates of growth, paulownia and balsa wood had superior ecologically characteristics compared to linden wood, being about five times higher [30,33]. From an ecological point of view, the results showed that paulownia and balsa wood are superior to linden wood but are very similar to each other, with the difference between balsa and paulownia of only 5.4%.

The quality coefficient (related to both compressive strength and modulus of resistance) of paulownia wood was better than balsa and linden. From this point of view, paulownia wood could be considered a strong species that is comparable to the strongest wood species in the word.

## 5. Conclusions

Generally speaking, the three deciduous woody species are as appropriate as possible, having well-observed similarities, and, because that, they can be considered homogenous.

The closest to linden is paulownia wood, which can replace linden in its uses in modeling, restoration, conservation, sculpture, etc.

Ecologically, balsa wood and paulownia wood are better than linden wood due to their faster growth.

The mechanical properties (compressive strength, MOR, MOE, and Brinell hardness) of the three species are consistent with their density.

From the point of view of quality coefficient, paulownia wood is the best species, compared to linden and balsa woods.

In a broader context, this research study falls into the area of knowledge of wood species with a low density and a good structural homogeneity.

Among the limitations of this research study and possible new areas of investigation there are the behavior of these species during thermal treatments at temperatures above 200 °C, in a nitrogen environment, or when there is no access to oxygen.

**Author Contributions:** Conceptualization, A.L. and A.A.; methodology, C.S.; software, C.S.I.; validation, A.L., A.A. and C.S.; formal analysis, C.S.I.; investigation, A.A.; resources, A.A.; data curation, A.L.; writing—original draft preparation, A.A.; writing—review and editing, AL.; visualization, C.S.I.; supervision, C.S.; project administration, A.L.; funding acquisition, A.L. All authors have read and agreed to the published version of the manuscript.

**Funding:** This research received no external funding.

**Institutional Review Board Statement:** Not applicable.

**Informed Consent Statement:** Not applicable.

**Data Availability Statement:** Not applicable.

**Acknowledgments:** The authors thank the Transilvania University of Brasov for the technical and logistical support provided.

**Conflicts of Interest:** The authors declare no conflict of interest.

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
