# Peer review of "Differences and Similarities between the Wood of Three Low-Density and Homogenous Species: Linden, Balsa, and Paulownia"

_applsci, doi:10.3390/app131810209_

Round 1

Reviewer 1 Report

Comments and Suggestions for Authors,

As I reviewed the manuscript “Some differences and similarities between wood of three low density and homogeneity species as linden, balsa and paulownia”, this manuscript presented a study on physical and mechanical properties of these species. I found that the overall of the study is clear. The manuscript needs to be revised because there is lack some information. Therefore, my response is "minor revision". Some minor requirements are needed as it seen in the comments.

1. I suggest reducing the keywords, not more than 5 words.

2. Figure 1, please scale in all sub pictures.

3. line 89, please use the same symbol (?0) in all content of manuscript.

4. Table 1, please use Linden instead of European Lime.

5. Table 2, What is the main point that author mentioned "Over Fiber Saturation Point" with different MC. Please clarify what information this table is supposed to communicate.

6.Table 4, please provide the SD with Mean.

7. It is recommended to add some comparisons to the literature data in the Results and Discussion section. A few studies are mentioned in the introduction that seem to indicate that in previous studies on another species with a lower density. Why did the authors select these species? What was different in the previous study?

Please recheck overall of grammatical errors.

Author Response

Reviewer 1

In this way, we thank the reviewer and his activity, because through the submitted work, through his suggestions and observations, he/she will certainly determine a larger audience among the readers of the paper and the journal in which this research is intended to be published.

  1. I suggest reducing the keywords, not more than 5 words.

Authors response: Only 5 keywords were kept.

  1. Figure 1, please scale in all sub pictures.

Authors response: In figures 1b and 1c, the measurement unit of 1 mm was arranged.

  1. line 89, please use the same symbol (?0) in all content of manuscript.

Authors response: It is the same symbol of density “ρ”, only that in the equation it is expressed by "Cambria Math" and in the text it is expressed by "Palatino Linotype". We cannot change the Greek symbol in the equation.

  1. Table 1, please use Linden instead of European Lime.

Authors response: The necessary change has been made, respectively “linden” word was introduced.

  1. Table 2, What is the main point that author mentioned "Over Fiber Saturation Point" with different MC. Please clarify what information this table is supposed to communicate.

Authors response: A paragraph was added to pages 9 and 10 that explained why the density above the saturation point of the fiber was chosen.

  1. Table 4, please provide the SD with Mean.

Authors response: A new row has been added at the end of the table, containing the standard deviation values.

  1. It is recommended to add some comparisons to the literature data in the Results and Discussion section. A few studies are mentioned in the introduction that seem to indicate that in previous studies on another species with a lower density. Why did the authors select these species? What was different in the previous study?

Authors response: - In the direction of the reviewer's suggestions, an addition was made to pages 9-10 in the results chapter, and three different additions were made to page 20 in the discussion chapter.

 -In the objectives part, a sentence was introduced to better explain these ambiguities. The two collaborators, Anamaria AVRAM and Constantin IONESCU, are specialized in the field of restoration-consolidation of wooden supports and want to see if paulownia wood can be used successfully in this field. It is intended to replace the balsa wood which is too weak and the linden wood which is too dense and hard (adds additional tension and weight to the restored-consolidated heritage objects). This work is intended to be a first step in this field.

  1. Please recheck overall of grammatical errors.

Authors response: All the work has been verified from a linguistic point of view.

Authors

Reviewer 2 Report

I have read the Manuscript and referred to some suggestions and comments. Thanks to the authors for doing a good study. The work will be more suitable for readers if the comments included.

11.       Keywords should generally contain words that reflect the content of the abstract and summarize the main theme, topic and key points of the article. So I advise you to provide a more comprehensive abstarct by incorporating your keywords into the abstract or alternatively, updating your current abstract to include the keywords.

2.       Line 31-36 this sentence too long and hard to understand. So please write short sentences.

3.       Line 72 and 73- you say that ………………….of the 3 woody species……………….of the 3 deciduous species……………. But Line 80 ………………….mass of the three wood species………………….  In my opinion, it would be more appropriate to write the number three as a word, but if you prefer to use the numbers, you should consistently indicate it throughout the entire article.

4.       Line 88 and 89 different symbolized the  P0. Please show the same symbol.

5.       You should refer every equation in the text.

6.       I suggest deleting the values above the error bars in the figures in the result section. Because it can be understood as standard deviation or it can make it difficult to understand the figures.

7.       Line 148- It should be shown as 3 superscripts and the rest of the text should be checked.

8.       Line 253- Physical and mechanical wood properties [1]  - [1]: what does this mean? Is it a reference?

9.       Were samples conditioned prior to mechanical testing? What is the moisture content of the test samples? You should refer to text

10.   You should calculate the total color differences (ΔE*) and compare to previous studies.

11.   Line -451 “table 7” - first letter must start with a capital letter

12.   References which have academic articles should be used instead of the number one reference (Wood Database.)

13.   The conclusion section of your paper should reflect not only the main findings but also the broader implications of your work. Make sure to include a discussion on the limitations of your study and suggest potential directions for future research.

Your manuscript is well-written. 

Author Response

Reviewer 2

In this way, we thank the reviewer and his activity, because through the submitted work, through his suggestions and observations, he will certainly determine a larger audience among the readers of the paper and the journal in which this research is intended to be published.

  1. Keywords should generally contain words that reflect the content of the abstract and summarize the main theme, topic and key points of the article. So, I advise you to provide a more comprehensive abstract by incorporating your keywords into the abstract or alternatively, updating your current abstract to include the keywords.

Authors response: The abstract has been adapted to the keywords. Also, based on another reviewer's observation, the keywords of the paper were reduced to 5.

  1. Line 31-36 this sentence too long and hard to understand. So please write short sentences.

Authors response: This long sentence was divided into two simple sentences. Also, each sentence has been significantly simplified.

  1. Line 72 and 73- you say that ………………….of the 3 woody species……………….of the 3 deciduous species……………. But Line 80 ………………….mass of the three wood species………………….  In my opinion, it would be more appropriate to write the number three as a word, but if you prefer to use the numbers, you should consistently indicate it throughout the entire article.

Authors response: All the numbers in the entire work have been changed into words.

  1. Line 88 and 89 different symbolized the P0. Please show the same symbol.

Authors response: It is the same density symbol from the Greek alphabet, but when converted to pdf, or due to the use of different fonts (Cambria Math and Palatino Linotype) they appear as different. In reality it is the same character.

  1. You should refer every equation in the text.

Authors response: All equation were referred in the text.

  1. I suggest deleting the values above the error bars in the figures in the result section. Because it can be understood as standard deviation or it can make it difficult to understand the figures.

Authors response: All values from specifically figures were erased with option “Data label” of Microsoft Excel.

  1. Line 148- It should be shown as 3 superscripts and the rest of the text should be checked.

Authors response: We changed it and the entire text were verified.

  1. Line 253- Physical and mechanical wood properties [1]  - [1]: what does this mean? Is it a reference?

Authors response: We added an expression on this line, in order to be more understanding the content. 

  1. Were samples conditioned prior to mechanical testing? What is the moisture content of the test samples? You should refer to text.

Authors response: Prior to mechanical testing the samples were conditioned, in order to obtain a moisture content of 12%. See line 176.

  1. You should calculate the total color differences (ΔE*) and compare to previous studies.

Authors responses: We also thought of calculating this difference (as the radical of the sum of the squares of the three parameters L, a, b), but the analyses would have been very complicated, because we have three woody species. Usually, this difference is determined when there are only two values to compare. Otherwise, the result obtained would not have been significantly different.

 Line -451 “table 7” - first letter must start with a capital letter

Authors responses: We changed it.

  1. References which have academic articles should be used instead of the number one reference (Wood Database)

Authors response: We introduced the reference "Wood database" because it contains most of the comparison data with the values of our research, especially for the analysed species. All other references usually contain only one comparison element (compression resistance, hardness, density, etc.). That is why we insist on keeping this reference, along with the others, even if it does not rise to a high academic level.

  1. The conclusion section of your paper should reflect not only the main findings but also the broader implications of your work. Make sure to include a discussion on the limitations of your study and suggest potential directions for future research.

Authors response: The conclusions chapter was completed according to the reviewer's suggestions.

 Authors,